# Diagnosis of Sarcopenia and Myosteatosis by Computed Tomography in Patients with Esophagogastric and Pancreatic Cancer

**DOI:** 10.3390/cancers16152738

**Published:** 2024-08-01

**Authors:** Nerea Sales-Balaguer, Patricia Sorribes-Carreras, Virginia Morillo Macias

**Affiliations:** 1Castelló Provincial Hospital Consortium Foundation, 12002 Castelló de la Plana, Spain; 2Nutrition and Dietetics Unit, Provincial Hospital Consortium of Castelló (CHPC), 12002 Castelló de la Plana, Spain; patsorribes@yahoo.es; 3Radiotherapy Service, Provincial Hospital Consortium of Castelló (CHPC), 12002 Castelló de la Plana, Spain

**Keywords:** sarcopenia, myosteatosis, computed tomography, cancer, malnutrition

## Abstract

**Simple Summary:**

Sarcopenia in cancer patients, understood as the progressive loss of muscle mass and function, reduces physical function, quality of life, and survival, and increases surgical complications and disease progression. Myosteatosis is an excess of fat mass infiltrated into the muscle, which causes a reduction in its functionality. Given the current scientific evidence on its reversibility, it is important to know the implications that its prevention, diagnosis, and early treatment could entail in the evolutionary course of the disease. Therefore, this study aims to make an early diagnosis and evaluate the incidence of sarcopenia and myosteatosis in patients diagnosed with esophageal, stomach, or pancreatic cancer and analyze their clinical evolution. In our sample, 22.2% of the patients had sarcopenia, and 60% had myosteatosis. At 6 months, 60% of the patients with sarcopenia died, compared with 14.3% of the patients who did not have sarcopenia. Therefore, it is important to make a correct early diagnosis to carry out a comprehensive approach that can reverse the patient’s situation. Likewise, prevention and care campaigns for muscle health could be carried out at all stages of life.

**Abstract:**

The increase in the global incidence of cancer highlights the need to continue advancing in the techniques of diagnosis and nutritional assessment of cancer patients, given the prognostic and therapeutic impact of nutritional status. In this study, sarcopenia was evaluated as an independent predictor of morbidity and mortality. Data from 45 patients diagnosed with esophagogastric or pancreatic cancer were analyzed. Body composition was determined using computed tomography images, and functionality tests were performed. Sarcopenia was present in 22.2% of the patients, while only 31.1% had correct musculature. A reduction in muscle mass or function was observed in 46.7% of the patients. Likewise, the prevalence of myosteatosis reached 60% of the patients. No significant differences were found with regard to the presence of sarcopenia according to BMI classifications, so it is necessary to evaluate the patient with body composition techniques that include the evaluation of the different muscle and fat compartments. In conclusion, a comprehensive intervention is necessary to improve the detection of sarcopenia/myosteatosis and, in the future, to be able to carry out an approach that improves the quality of life and survival rates of patients.

## 1. Introduction

Cancer is one of the leading causes of mortality and morbidity worldwide, and an increase in incidence is expected in the coming decades. At the same time, its treatment (surgery, radiotherapy, and different pharmacological therapies) advances in sophistication and precision to address the specific characteristics of the individual. Although many tumors cannot be cured yet, they can become chronic diseases, so it is necessary to advance in diagnostic techniques and nutritional assessment [1].

Musculoskeletal mass has important functions beyond locomotion. Its ability to uptake glucose stimulated by insulin, its influence on bone density through mechanical force on bones, and its impact on protein metabolism throughout the body stand out [2,3,4,5].

Sarcopenia is the loss of muscle mass (MM) and skeletal function quantified by objective measures of muscle mass, strength, and physical function [2,6,7,8,9,10,11,12]. Presarcopenia can be defined as low muscle mass with no loss of strength or disability or low physical performance [13]. The prevalence of sarcopenia ranges from 16% to 71%, depending on the definition and the type of tumor considered in the study [8,9,14,15,16,17].

Likewise, age, gender, and level of physical activity are some of the risk factors identified. This pathology is closely related to malnutrition, including deficient protein and energy intake, immobility/sedentary lifestyle, and systemic diseases, highlighting inflammation (especially low grade) as a key factor in the pathophysiological process [18,19,20]. In it, there are various pathophysiological processes, such as mitochondrial dysfunction and inflammatory and hormonal changes, which cause adverse effects due to the reduction in lean body mass: increased falls, functional deterioration, frailty, and mortality [19,21,22,23,24,25].

Medical treatments can be hindered by the development of malnutrition and metabolic alterations present, induced either by the tumor itself, by its treatment, or by the symptoms that the patient may present. Malnutrition and loss of muscle mass in this population is common and has a negative impact on clinical outcomes. Its causes can be multifactorial: inadequate food intake, reduced physical activity, catabolic states, etc. Therefore, screening, prevention, early intervention, and monitoring of malnutrition with advanced techniques that can detect changes are important [1,26].

On the other hand, myosteatosis is the excess of fat mass infiltrated into the muscle, which causes a reduction in its functionality. It is considered a marker of muscle quality. In other words, a muscle with a higher fat deposit has less contraction power and ability to produce force per muscle unit. The causes of ectopic fat deposition are multifactorial: hormonal alterations, capillary blood flow, age, physical inactivity, obesity, and mitochondrial abnormalities. In addition, the alteration of metabolism involving increased lipolysis induced by inflammation, hyperlipidemia, and/or progressive lipid redistribution to newly formed visceral fat deposits and nearby muscles promotes fat infiltration into the muscle [27]. Excess lipids are thought to enter the skeletal muscle, which accumulates them in the form of intramuscular adipose tissue (TAIM), intermuscular, and intramyocellular lipid droplets or fatty acid derivatives. It is an abnormal phenomenon that increases with aging and is negatively correlated with muscle mass, strength, mobility, and the proper functioning of metabolism [27].

Dynapenia is the loss of muscle strength. The grip strength of the hand has been validated for its diagnosis, being predictive of long-term mortality in the general population [28]. A reduction in strength can be explained by a decrease in muscle mass, but also by other factors, such as the presence of myosteatosis [27]. It has been considered that pretherapeutic value and changes in grip strength during chemotherapy can predict toxicity, particularly neurotoxicity. This would involve dose adjustments, delays, or interruptions of treatment [28].

Obesity and sarcopenia can coexist in the form of sarcopenic obesity (OS), so sedentary lifestyles pose additional challenges [18,29,30,31]. Major international societies point to obesity along with altered body composition as a clinical and scientific priority for practitioners [30]. OS is defined as the coexistence of excess adiposity and low muscle mass/function. Its diagnosis should be considered in people who have a positive screening test and a high body mass index or waist circumference and markers of low muscle mass or function (risk factors, clinical symptoms, or through validated questionnaires) [31]. These people may appear healthy due to the masking effect of both pathologies. Age-related obesity and muscle atrophy are closely related and reciprocally regulated by adipose tissue and skeletal muscle dysfunction. During aging, adipose inflammation leads to the redistribution of fat to the intra-abdominal area (visceral fat) and fatty infiltrations into skeletal muscles, leading to decreased strength and functionality. A long period of systemic inflammation would lead to the atrophy and wasting seen in entities such as cachexia and sarcopenia [15,32,33,34].

There is increasing scientific evidence for the estimation of body composition. It is proposed that traditional anthropometric parameters are insufficient to estimate body compartments. Its analysis involves the qualitative and quantitative measurement of the different types of tissue that make up the human body [29,35,36,37,38]. The evaluation of composition by means of diagnostic images is a quantitative practice with great scientific impact and is used more and more frequently, with more precise results and clinical implications of greater relevance. Through this practice, visceral adipose tissue (TAV) and MM (compartments associated with pro-inflammatory activity, which has important repercussions on the risk of the appearance of associated diseases, such as diabetes, cardiovascular diseases, and increased oncological risk) can be accurately estimated [16,29,39,40,41].

To this end, one of the most precise techniques is axial slicing by computed tomography (CT) at the level of the third lumbar vertebra, validated by various societies and scientific articles [4,39]. However, it is necessary to have specific software that helps to quantify muscle mass, as well as subcutaneous fat (TAS), intramuscular and visceral. Usually, this way of segmenting images is a very laborious and arduous task [4,40,41], and software usually needs specific licenses [4,42]. Segmentation is an image processing term that refers to the labeling of each individual pixel (or voxel, which consists of three-dimensional pixels) according to tissue or organ. Since CT and MRI are frequently used in clinical practice, segmentation allows the opportunistic assessment of body composition in clinical cohorts without the need for further studies [43]. 

Therefore, malnutrition and loss of muscle mass are associated with a reduction in physical function, reduction in quality of life, dose-limiting toxicity, reduced response to treatment, increased risk of surgical complications, and reduced survival of cancer patients [19,35,44,45,46]. Sarcopenia (both low muscle mass and reduced muscle density) has been reported to be an independent predictor of survival, especially in patients with gastrointestinal, respiratory, and urothelial cancer. Sarcopenia is a strong predictor of increased postoperative complications and a robust prognostic factor for therapeutic responses and outcomes [47]. Prior knowledge of the therapeutic approach can improve the patient’s clinical outcomes, thanks to early intervention [14,17,23,37].

## 2. Objective of the Study

### 2.1. Primary

To determine body composition and diagnose presarcopenia/sarcopenia and myosteatosis, in patients with pancreatic, esophageal, and stomach cancer, using computed tomography and other methods, such as anthropometry, biochemistry, and dynamometry, as the main working tool.

### 2.2. Secondary

To determine the prevalence of excess body fat by quantifying visceral, subcutaneous, and intramuscular adipose tissue in our study population or in that subgroup of patients;To analyze the correlation between the muscle mass determined at the L3 level in patients who have a computed tomography with both levels included and the cut-off points established as a reference between normality and sarcopenia.

## 3. Methods

This is a cross-sectional, observational, and descriptive study in which a detailed nutritional evaluation is carried out on patients diagnosed with pancreatic, esophageal, or stomach cancer who are referred to the Nutrition and Dietetics Unit of the Provincial Hospital Consortium of Castellón (CHPC). This study has been accepted by the Research Ethics Committee of the Provincial Hospital Consortium of Castellón on 22 June 2022 (FHPCS-22-003).

Inclusion criteria

Patients over 18 years of age with a previously mentioned diagnosis;Patients who have a CT scan image;Patients who are going to receive cancer treatment.

Exclusion criteria

Pregnant women;Pacemaker or defibrillator carriers;Patients who have poor-quality CT images (incomplete, blurry, cropped, or having too much contrast);Wearers of metal prostheses.

Population studied

A total of 45 patients diagnosed with esophageal, stomach, or pancreatic cancer who were referred to the CHPC’s Nutrition and Dietetics Unit from January to December 2023 were analyzed.

Statistical analysis

The Jamovi database was used, with R as the underlying infrastructure. To this end, the frequency and descriptive tools r-Student, chi-square test, and ANOVA were used.



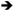

Analysis of muscle functionality by hand grip force and gait test

Its analysis is carried out by the grip force of the hand using a dynamometer for both the right and left arm. We performed three determinations in each hand with the sitting, unsupported subject on the arm under study using a JAMAR (Hydraulic Hand Dynamometer, J. A. Preston Corporation, Clifton, NJ, USA) hand dynamometer. Finally, the mean was calculated, and a decreased strength or dynapenia was considered if the value was <27 kg/m^2^ in men and <16 kg/m^2^ in women.

In addition, the functionality was also measured using the Timed Up and Go (TUG) test. The individual was asked to get up from a chair, walk to a marker located 3 m away, turn around, walk again, and sit down. The cut-off point was set above 20 s.



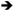

Determination of body composition in the third lumbar vertebra (L3) from computed tomography

A single cross section analyzes the body composition (MM, TAIM, TAV, SAP) in the third lumbar vertebra, according to different Hounsfield units (HU), being −29 +150 for MM, −190 −30 for TAS/TAIM, and −150 to −50 for TAV.

The MM (cm^2^) is automatically calculated by adding pixels of tissue and dividing by the volume of the set cut. The muscles delineated for the musculoskeletal index (SMI) are the transversus abdominis muscle, external oblique, internal oblique, psoas, rectus abdominis, quadratus lumborum, and the erector spinatus muscle. Skeletal muscle mass is calculated at the L3 level, with the total transverse skeletal muscle area normalized by height divided by height squared. In this way, two different cut-off points of the musculoskeletal index specific to sex and body mass index (BMI) have been established. The established cut-off points for the diagnosis of decreased muscle mass are lumbar MSI ≤ 41 cm^2^/m^2^ for women and ≤53 cm^2^/m^2^ for men, with BMI ≥ 25 kg/m^2^ and ≤43 cm^2^/m^2^ for men and BMI < 25 kg/m^2^ for women. Muscle density is determined according to the UH of the MME determined at L3. The established cut-off points for low muscle density or myosteatosis are < 33 HU for females and <41 HU for males.

In view of the results, we considered it convenient to evaluate the fat compartments, since we do not have cut-off points. There is a need to evaluate the body composition in its entirety and not just the compartment of the muscles, since the body is interconnected, and the adipose tissue is also involved in the patient’s metabolic processes. Specifically, we evaluated visceral, intramuscular, and subcutaneous adipose tissue. The TAV, TAS, and TAIM are determined at the L3 level. For its evaluation, the pixels of each fabric were added again and divided by the established cutting volume. Muscle density was also determined according to the UH of the IME. The lower the density, the greater the fat infiltration within the muscles. The cut-off points established for myosteatosis are < 33 for women and <41 for men.

A specific image analysis program (3D Slicer (version 5.6.1), Brigham and Women’s Hospital, Georgia Institute of Technology, Harvard University, Boston, MA, USA) was used for evaluation. It is a free cross-platform software application for the computation of medical images. This software also provides advanced functionalities, such as automated segmentation and registration for a variety of application domains. It is a multi-platform software that can directly visualize the anatomy of patients and merge anatomical data and functions for multimodal processing and analysis.



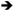

Classification of sarcopenia/presarcopenia/myosteatosis

Three categories were established—presarcopenia, sarcopenia, and nonsarcopenia—depending on the presence of low muscle mass and the presence or absence of functional impairment. Two categories were also established according to the presence or absence of myosteatosis.

## 4. Results

According to the aforementioned criteria, the body composition of our population was detected. Our sample had 13 patients with esophageal cancer, 15 with stomach cancer, and 17 with pancreatic cancer at stages 3–4. Only one of each group had stages 1–2. A sample prevalence of sarcopenia of 22.2% was observed. Likewise, patients who had correct muscle mass and functionality were only 31.1%. However, it is important to note that 46.7% of the patients showed a reduction in muscle mass or grip strength (presarcopenia).

No significant differences were found in the presence or not of sarcopenia, myosteatosis, dynapenia, or sarcopenic obesity according to the location of the tumor, so all patients were analyzed as a whole.

### 4.1. Sarcopenia

The presence or absence of sarcopenia does not have a correlation with body mass index, so this indicates that it is necessary to develop advanced assessment techniques that allow for determining body composition beyond those traditionally used. Therefore, patients with sarcopenia may have body mass indexes that indicate underweight, normal weight, overweight, and obesity, together with low muscle function (Figure 1).

Specifically, the mean body compositions of patients with sarcopenia, presarcopenia, and nonsarcopenia are shown in Table 1.

Figure 2 shows the differences in musculoskeletal index and grip strength in the groups of patients with sarcopenia, presarcopenia, and nonsarcopenia (*p* < 0.01).

On the other hand, the SARC-F screening tool was also evaluated, showing a significant correlation predictor of sarcopenia (*p* < 0.001). This shows the importance of implementing screening tools that detect the risk of suffering from sarcopenia early.

### 4.2. Sarcopenic Obesity

Four patients had sarcopenic obesity (8.9%). That is, they had a BMI > 30 kg/m^2^; however, the calculated muscle mass was below range.

### 4.3. Myosteatosis

On the one hand, 60% of the patients had a high level of fat infiltration in the muscle (Figure 3). The presence or absence of myosteatosis was not significantly correlated with functional tests.

On the other hand, it was not correlated with the musculoskeletal index, nor did it show differences in relation to fat compartments.

### 4.4. Fatty Compartments

Figure 4 and Figure 5 show the differences in fat compartments according to BMI. Significant differences are shown in visceral and subcutaneous adipose tissue according to the patient’s BMI.

There are differences in visceral and subcutaneous (but not intramuscular) adipose tissue according to the presence of overweight and obesity. Therefore, 150 cm^2^/m^2^ for the TAV and 120 cm^2^/m^2^ for the TAS are proposed as cut-off points. More studies will be needed to support these results.

### 4.5. Muscle Functionality

Dynapenia was present in 55.6% of the patients, which has no correlation with BMI, but does have with muscle density (*p* < 0.5). Of the patients, 33.3% had dynapenia without low muscle mass.



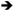

Mortality

After 6 months, 60% of the patients with sarcopenia died, compared with 14.3% of the patients who did not have sarcopenia. Patients classified as having presarcopenia had a mortality rate of 57.1% (*p* < 0.05). HR, 95%, for survival analysis was also calculated (Table 2).

There were no differences in patient mortality according to dynapenia or myosteatosis.

## 5. Discussion

The overall rates of sarcopenia in the clinical setting range from 5% to 25% of patients admitted to medical/surgical units, 60% to 70% of critical patients, and 15% to 50% of cancer patients [6,48,49]. In oncology, depending on the definition and location of the tumor, the prevalence is between 16% and 71%. In locally advanced esophageal cancer, its prevalence is 16% at diagnosis and 31% after adjuvant therapy and 35% in survivors after 1 year. Our data are like those reported in the literature, although we have not found studies that reflect the prevalence of presarcopenia in these patients. In our study, half of the sample showed a reduction in functionality or MM. These data emphasize the importance of an early approach, which can reverse the situation of presarcopenia in most patients before starting the therapeutic approach [6,35,48]. 

Numerous studies indicate the importance of body composition analysis in cancer patients, but muscle density and fat compartments are often overlooked. This study looks at muscle fat infiltration, visceral fat, subcutaneous fat, and muscle mass, as well as their density and functionality as a whole body and not just separate compartments.

On the other hand, more studies would be needed to support the cut-off points to evaluate the excess of the different fat compartments.

The detection of sarcopenic obesity is important in patients, given that its metabolic implications are high, as well as the difficulty of detecting them without more sophisticated techniques. As our research has seen, it is important to use validated and sensitive screening tools that help quickly detect the risk of suffering from sarcopenia.

In addition, the introduction into the clinical practice of measuring muscle functionality helps to assess the patient’s evolution and detect a reduction in grip strength. Considering the results obtained, muscle functionality could be altered by the decrease in muscle mass and the reduction in muscle density and fat infiltration. Since myosteatosis has not been correlated with functional tests, the correct grip strength result or TUG is not predictive of the presence or absence of myosteatosis.

As regards the software used, it was 3D Slicer (Brigham and Women’s Hospital, Georgia Institute of Technology, Harvard University, Boston, MA, USA), open-sourced and validated by Blanc-Durand et al. [14,15,16,17]. Other similar tools that are also used for body composition analysis, such as OsiriX (Pismo SARL, Geneva, Switzerland), BioImage Suite (Yale University, New Haven, CT, USA), MIPAV (National Institute of Biomedical Imaging and Bioengineering (NIBIB), part of the National Institutes of Health (NIH), Bethesda, MD, USA), and ImageJ (NIH, Bethesda, MD, USA), provide extensible development platforms for biomedical imaging applications [16,50]. In this case, the system is available in a free, open-source version or as a commercial product authorized by the Food and Drug Administration (FDA) [51,52]. One drawback to its use is its dependence on operating systems and the cost of the product under restrictive licenses for an open source. Moreover, it is not an FDA-approved device, and its license makes no claim about the clinical applicability of the software; however, it has been applied in a variety of projects that have appropriate research oversight ensuring authorized use in research [51,52,53].

The use of CT scan data to assess body composition has expanded understanding about the relationship between muscle mass and treatment tolerance for cancer, complications, and survival [19,38,39,54,55].

These results, in particular, highlight the importance of addressing muscle loss from a comprehensive approach, highlighting the importance of adopting a multidisciplinary approach, considering the early nutritional support necessary in each individual case, correct and adapted physical exercise, and medical treatment and establishing and promoting psychosocial resilience (Figure 6) [44]. In subsequent studies, it would be interesting to evaluate the oncological impact after the nutritional approach [22,24,54].

Likewise, physical exercise combined with correct nutritional support is a particularly effective intervention to mitigate muscle loss and induce protein anabolism [56].

## 6. Conclusions

There is a high prevalence of sarcopenia and presarcopenia in the patients studied, as well as myosteatosis. Further research is needed, in both the assessment of muscle mass and fat distribution and the optimal and comprehensive approach that can prevent and/or reverse sarcopenia in patients with the aim of improving quality of life and survival rates in this population.

## 7. Strengths and Weaknesses

The weaknesses of our study would be the small sample size and the participation of a single center. However, in terms of strengths, we highlight of the study of muscle mass and fat mass in combination and originality.

## Figures and Tables

**Figure 1 cancers-16-02738-f001:**
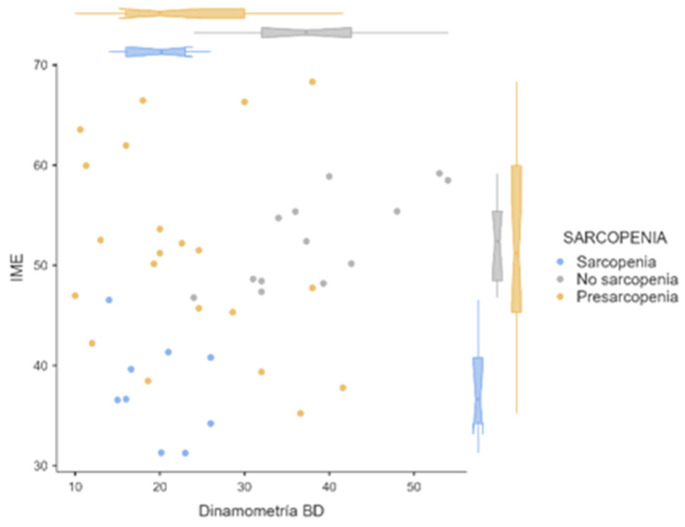
Distribution of the musculoskeletal index and dynamometry according to sarcopenia, nonsarcopenia, or presarcopenia. Blue: sarcopenia; gray: no sarcopenia; yellow: presarcopenia.

**Figure 2 cancers-16-02738-f002:**
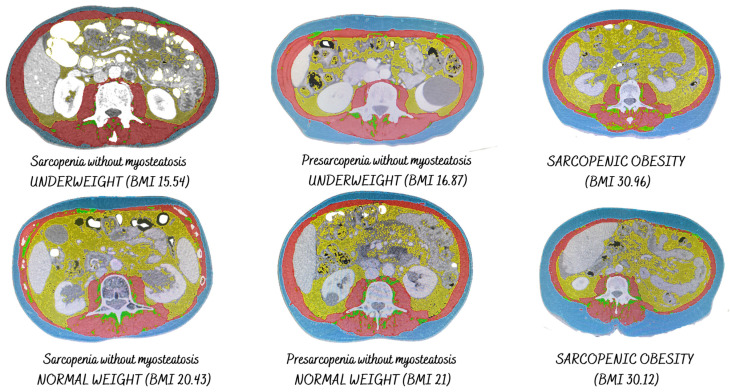
Presence of sarcopenia according to BMI.

**Figure 3 cancers-16-02738-f003:**
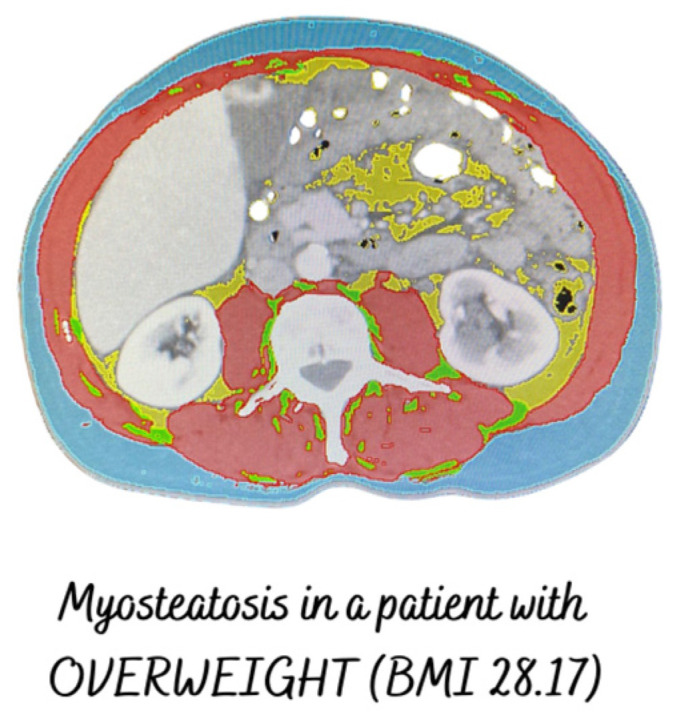
Patient with myosteatosis, muscle mass in range, and overweight.

**Figure 4 cancers-16-02738-f004:**
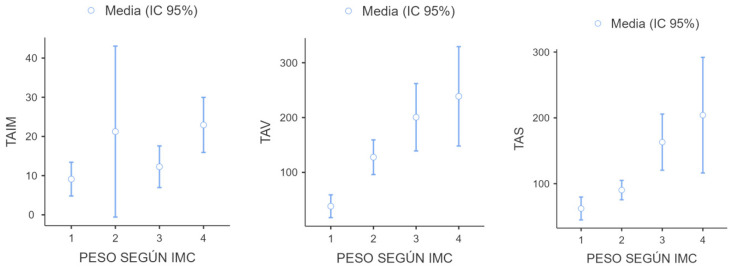
Value of fat compartments according to the patient’s BMI (cm^2^). 1: Low weight; 2: Normal weight; 3: overweight; 4: Obesity.

**Figure 5 cancers-16-02738-f005:**
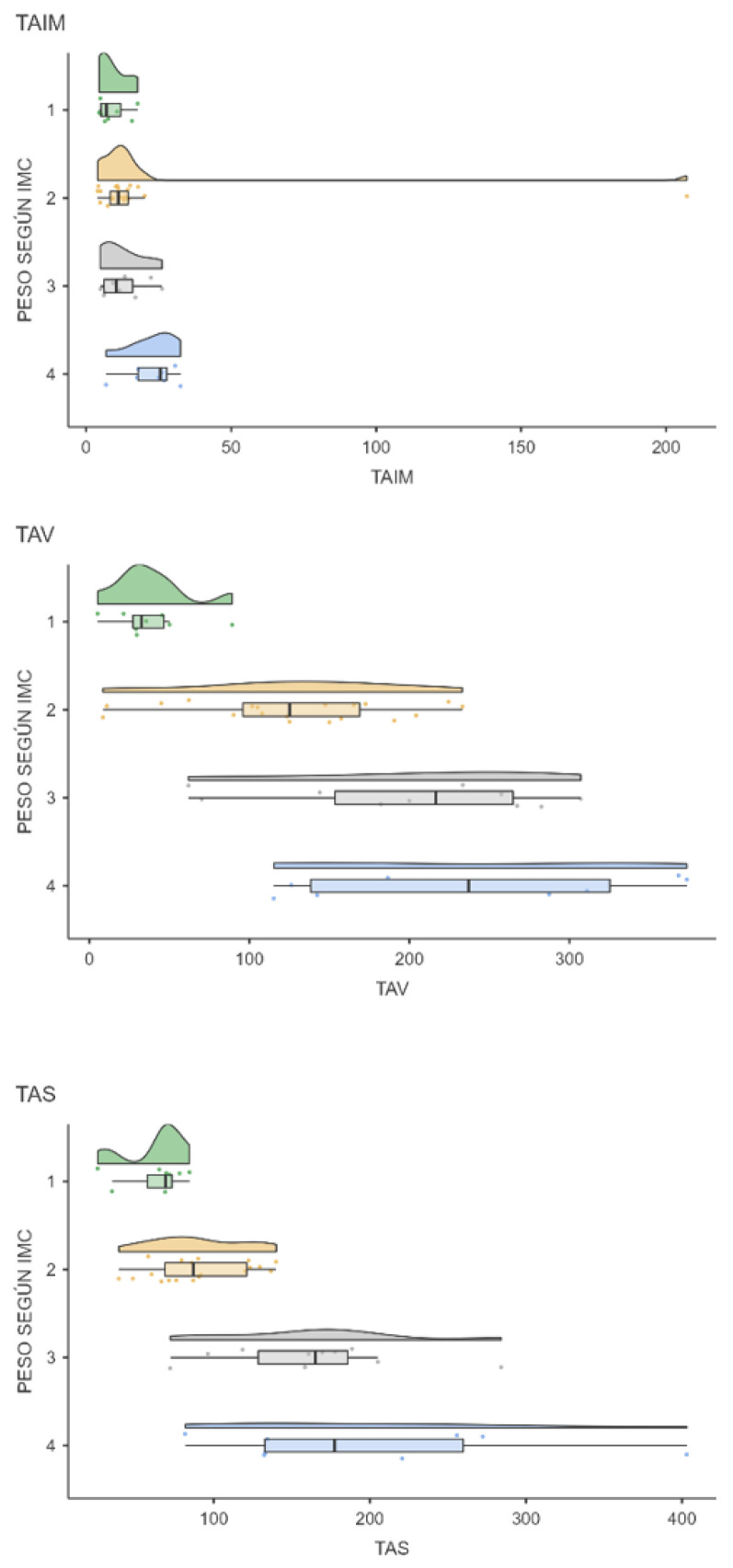
Distribution of TAIM, TAV, and TAS (cm^2^) according to the patient’s BMI: 1: low weight; 2: normal weight; 3: overweight; 4: obesity.

**Figure 6 cancers-16-02738-f006:**
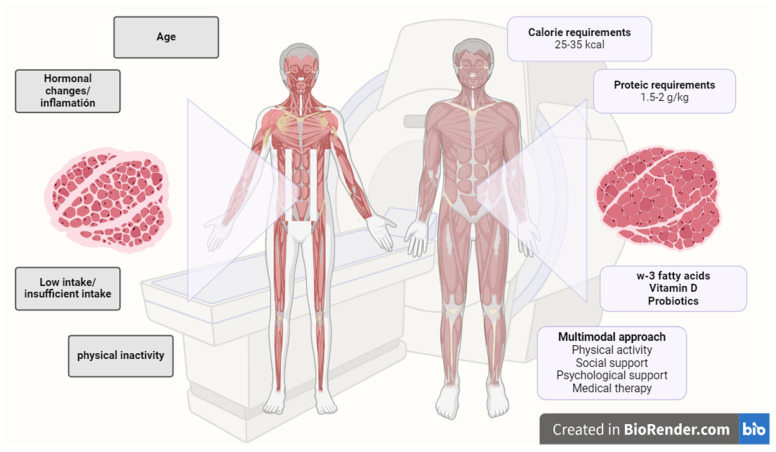
Needs for a correct evaluation and management of sarcopenia. Created with BioRender.

**Table 1 cancers-16-02738-t001:** Body composition according to sarcopenia, no sarcopenia, or presarcopenia. MM: muscle mass; TAIM: intramuscular adipose tissue; TAV: visceral adipose tissue; TAS: subcutaneous adipose tissue (cm^2^).

						Shapiro–Wilk	
	Sarcopenia	N	Average	Minimum	Maximum	W	*p*
MM	Sarcopenia	10	109.5	89.81	141.4	0.925	0.404
	No Sarcopenia	14	152.1	116.48	180.1	0.975	0.938
	Presarcopenia	21	130.8	96.38	167.8	0.946	0.280
TAIM	Sarcopenia	10	35.6	4.90	207.2	0.491	<0.01
	No Sarcopenia	14	10.2	3.93	30.7	0.798	0.005
	Presarcopenia	21	13.5	4.52	32.6	0.915	0.070
TAV	Sarcopenia	10	118.2	11.04	368.1	0.854	0.064
	No Sarcopenia	14	175.2	45.06	310.8	0.958	0.696
	Presarcopenia	21	143.4	5.20	373.3	0.954	0.408
TAS	Sarcopenia	10	117.4	47.99	220.7	0.883	0.142
	No Sarcopenia	14	116.7	39.21	188.7	0.968	0.856
	Presarcopenia	21	127.1	25.64	403.1	0.803	<0.01

**Table 2 cancers-16-02738-t002:** HR, 95%, for survival analysis.

HR, 95%	Esophagus	Stomach	Pancreas
Sarcopenia			
Nonsarcopenia	2.57 (0.62–10.61, *p* = 0.192)	2.53 (0.59–10.76, *p* = 209)	0.63 (0.11–3.53, *p* = 0.603)
Presarcopenia	0.63 (0.14–2.81, *p* = 0.545)	0.97 (0.24–3.968, *p* = 0.968)	1.42 (0.45–4.47, *p* = 0.546)

## Data Availability

The datasets presented in this article are not readily available because the data are part of an ongoing study and due to technical limitations. Requests to access the datasets should be directed to the authors.

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
