# Peer review of "Diagnosis of Sarcopenia and Myosteatosis by Computed Tomography in Patients with Esophagogastric and Pancreatic Cancer"

_cancers, 2024, doi:10.3390/cancers16152738_

Round 1

Reviewer 1 Report

Comments and Suggestions for Authors

Introduction

The authors should include data about the prevaloence of low skeletal muscle mass and myosteatosis in oncology and, in pancreatic and esophageal cancer.

Methods:

Study design?

Thresholds for sarcopenia and presarcopenia are not defined! It should be included!

Oncological setting??

Results

how many patients had esophageal/gastric cancer and how many pancreatic cancer?? It should be given. Tumor stages should be added.

Mortality: the association between sarcopenia and mortality should be presented as HR, 95%CI (regression analysis). It should be given for gastric and pancreatic cancer separately!

grip strength data are not presented!

Correlation between grip strength and CT body composition should be given.

Discussion

Novelty of the present report should be mentioned. There are already numerous large studies/meta analysies about the topic.

Clinical relevance?

Conclusion

too large and general. It should be short and should reflect the results of the present study.

Some relevant previous reports should be included:

doi: 10.1200/JCO.21.00102. 

doi: 10.1016/j.ejca.2023.112939. 

doi: 10.1016/j.clnu.2021.08.023.

doi: 10.1016/S0140-6736(19)31138-9.

doi: 10.1016/S1470-2045(08)70153-0. 

Comments on the Quality of English Language

Introduction

The authors should include data about the prevaloence of low skeletal muscle mass and myosteatosis in oncology and, in pancreatic and esophageal cancer.

Methods:

Study design?

Thresholds for sarcopenia and presarcopenia are not defined! It should be included!

Oncological setting??

Results

how many patients had esophageal/gastric cancer and how many pancreatic cancer?? It should be given. Tumor stages should be added.

Mortality: the association between sarcopenia and mortality should be presented as HR, 95%CI (regression analysis). It should be given for gastric and pancreatic cancer separately!

grip strength data are not presented!

Correlation between grip strength and CT body composition should be given.

Discussion

Novelty of the present report should be mentioned. There are already numerous large studies/meta analysies about the topic.

Clinical relevance?

Conclusion

too large and general. It should be short and should reflect the results of the present study.

Some relevant previous reports should be included:

doi: 10.1200/JCO.21.00102. 

doi: 10.1016/j.ejca.2023.112939. 

doi: 10.1016/j.clnu.2021.08.023.

doi: 10.1016/S0140-6736(19)31138-9.

doi: 10.1016/S1470-2045(08)70153-0. 

Author Response

Thank you so much for your replay and your help.

Comments: Introduction - The authors should include data about the prevaloence of low skeletal muscle mass and myosteatosis in oncology and, in pancreatic and esophageal cancer.

Response: added

Comments: - Methods:

Study design? --> This is a cross-sectional, observational and descriptive study

Thresholds for sarcopenia and presarcopenia are not defined! It should be included! --> Added

Oncological setting?? The oncological setting: all patients were in stage 3-4, exept 1 of each cancer type, that were in 1.

Comments- Results

how many patients had esophageal/gastric cancer and how many pancreatic cancer?? It should be given. Tumor stages should be added. --> added

Mortality: the association between sarcopenia and mortality should be presented as HR, 95%CI (regression analysis). It should be given for gastric and pancreatic cancer separately! --> added

grip strength data are not presented! --> Dynapenia was the lost of strenght (under the thresholds). So, the data from grip strenght and timed up and go were included in muscle funcionality like dynapenia. 

Comments: - Discussion

Novelty of the present report should be mentioned. --> added: There are already numerous large studies/meta analysies about the topic. Numerous studies indicate the importance of body composition analysis in the cancer patient, but muscle density and fat compartments are often overlooked. This study looks at muscle fat infiltration, visceral fat, subcutaneous fat and muscle mass, as well as their density and functionality as a whole body and not just separate compartments 

Comments: - Conclusion

too large and general. It should be short and should reflect the results of the present study.--> modified

Reviewer 2 Report

Comments and Suggestions for Authors

his is an interesting article. I have some suggestions:

  1. The relationship between malignancy and sarcopenia can be emphasized more. The following landmark article can be referenced: https://pubmed.ncbi.nlm.nih.gov/36307591/
  2. The number of IRB approval should be stated at the beginning of the methods section.
  3. I suggest having a separate paragraph for the definition of sarcopenia, with a pertinent subtitle.
  4. Has a sample size estimation been performed?

Author Response

Thank you so much for your replay and your help.

  1. The relationship between malignancy and sarcopenia can be emphasized more. The following landmark article can be referenced: https://pubmed.ncbi.nlm.nih.gov/36307591/ 

Response: added

2. The number of IRB approval should be stated at the beginning of the methods section.

Response: added

3. I suggest having a separate paragraph for the definition of sarcopenia, with a pertinent subtitle.

Response: now it is in a separate paragraph.

4. Has a sample size estimation been performed?

Response: An estimate of the sample size needed for the study has been made considering Spearman's correlation coefficient with a power of 80% and a significance of 0.05 to find the expected correlation coefficient of 0.3, requiring a total sample of 90 patients, considering a 5% loss of patients during the course of the study.